# Molecular Dynamics Simulation of Forsterite and Magnesite Mechanical Properties: Does Mineral Carbonation Reduce Comminution Energy?

**Akash Talapatra * and Bahareh Nojabaei**

Department of Mining and Minerals Engineering, Virginia Tech, Blacksburg, VA 24061, USA; baharehn@vt.edu
* Correspondence: takash@vt.edu

**Abstract:** This work compares the mechanical properties of two geomaterials: forsterite and magnesite. Various physical conditions are considered to investigate the evolution of stress–strain relationships for these two polycrystals. A molecular-scale study is performed on three-dimensional models of forsterite and magnesite. Three different temperatures (300 K, 500 K, and 700 K) and strain rates (0.001, 0.01, and 0.05 ps$^{-1}$) are considered to initiate deformation in the polycrystals under tensile and compressive forces. The polycrystalline structures face deformation at lower peaks at high temperatures. The Young's modulus values of forsterite and magnesite are found to be approximately 154.7451 GPa and 92.84 GPa under tensile forces and these values are found to be around 120.457 GPa (forsterite) and 77.04 GPa (magnesite) for compressive forces. Increasing temperature reduces the maximum strength of the polycrystalline structures, but forsterite shows higher ductility compared to magnesite. Strain rate sensitivity and the effect of grain size are also studied. The yield strengths of the forsterite and magnesite drop by 7.89% and 9.09% when the grain size is reduced by 20% and 15%, respectively. This study also focuses on the changes in elastic properties for different pressures and temperatures. In addition, from the radial distribution function (RDF) results, it was observed that the peak intensity of pairwise interaction of Si–O is higher than that of Mg–O. Finally, it is found that the formation of magnesite, which is the product of mineral carbonation of forsterite, is favorable in terms of mechanical properties for the comminution process.

**Keywords:** mineral carbonation; comminution energy; stress–strain relationship; elastic properties; radial distribution function

## 1. Introduction

Carbon mineralization is an emerging approach to storing $CO_2$ in the form of carbonate minerals, particularly in calcium, magnesium, and silicate-rich rocks/geomaterials such as olivine, wollastonite, and serpentine. This process occurs naturally during the weathering of these rocks/geomaterials [1,2]. The major pathways and kinetics of storing enriched $CO_2$ in carbonate minerals have been described frequently in the literature to discuss the potential and required cost of this process. Researchers have not only focused on the potential of carbon mineralization in minimizing greenhouse gas emissions and assuring storage is non-toxic and permanent, but they have also been searching for emulating and accelerating the spontaneity as well as the balance of this process within the Earth's deep interior [3–5].

Mineral carbonation is a process of reacting carbon dioxide ($CO_2$) with alkaline and alkaline earth-bearing (magnesium and silicate-rich) minerals to form stable carbonate minerals [6]. A variety of silicate mineral groups containing $Ca^{2+}$, $Mg^{2+}$, and $Fe^{2+}$ ions present in nature for targeting mineral carbonation are olivine, serpentine, pyroxene, mica group, and clay minerals. However, past work has claimed that olivine-group minerals, particularly forsterite ($Mg_2SiO_4$), are the best potential feedstock for the carbon mineralization process and form stable carbonate minerals ($MgCO_3$). Forsterite is abundant in

the Earth's crust and comprises the upper mantle. It is a common mineral in ultramafic rocks and formed as a result of the cooling and solidification of magma and it is much more stable at high temperatures and pressure. Furthermore, forsterite has a high surface area-to-volume ratio, making it highly reactive with $CO_2$ [7,8]. It has a $CO_2$ sequestration potential of around 2014.7–1896.3 kg/m$^3$ (Table 1). Magnesite, on the other hand, is a carbonate-rich mineral found in tectonically active regions and comprises the Earth's lower mantle and is less stable at higher temperatures and pressure. Figure 1 shows a schematic view of the 3D atomic and crystalline structures of forsterite and magnesite. Both forsterite and magnesite are polycrystals, with forsterite crystalizing in orthorhombic systems and magnesite crystalizing in cubic systems. A consideration of crystal orientation and lattice parameters is provided in the next section.

**Table 1.** $CO_2$ sequestration potential of major rock-forming minerals [9].

| Mineral Name | Formula | Potential $CO_2$ Fixed, kg/m$^3$ Mineral |
| --- | --- | --- |
| Olivine group (forsterite) | $Mg_2SiO_4$-$Fe_2SiO_4$ | 2014.7–1896.3 |
| Pyroxene group (enstatite) | $(Mg, Fe)_2 Si_2O_6$ | 1404 |
| Serpentine | $Mg_3Si_2O_5(OH)_4$ | 1232 |
| Wollastonite | $CaSiO_3$ | 1097.1 |
| Amphibole group (hornblende) | $Ca_2Na_{0-1}(Mg,Fe(II)_{3-5}(Al,Fe(III)_{2-0}[Si_{6-8}Al_{2-0}O_{22}](O,OH)_2$ | 1000.4 |
| Mica group (biotite) | $K_2(Mg,Fe(II))_{6-4}(Fe(III),Al)_{0-2}[Si_{6-5}Al_{2-3}O_{20}](OH)_{4-2}$ | 671 |
| Plagioclase (anorthite) | $Ca[Al_2Si_2O_8]$ | 436.4 |
| Clay Minerals (smectite) | $(1/2Ca,Na)_{0.7}(Al,Mg,Fe)_4(Si,Al)_8O_{20}(OH)_4.nH_2O$ | 161.2 |

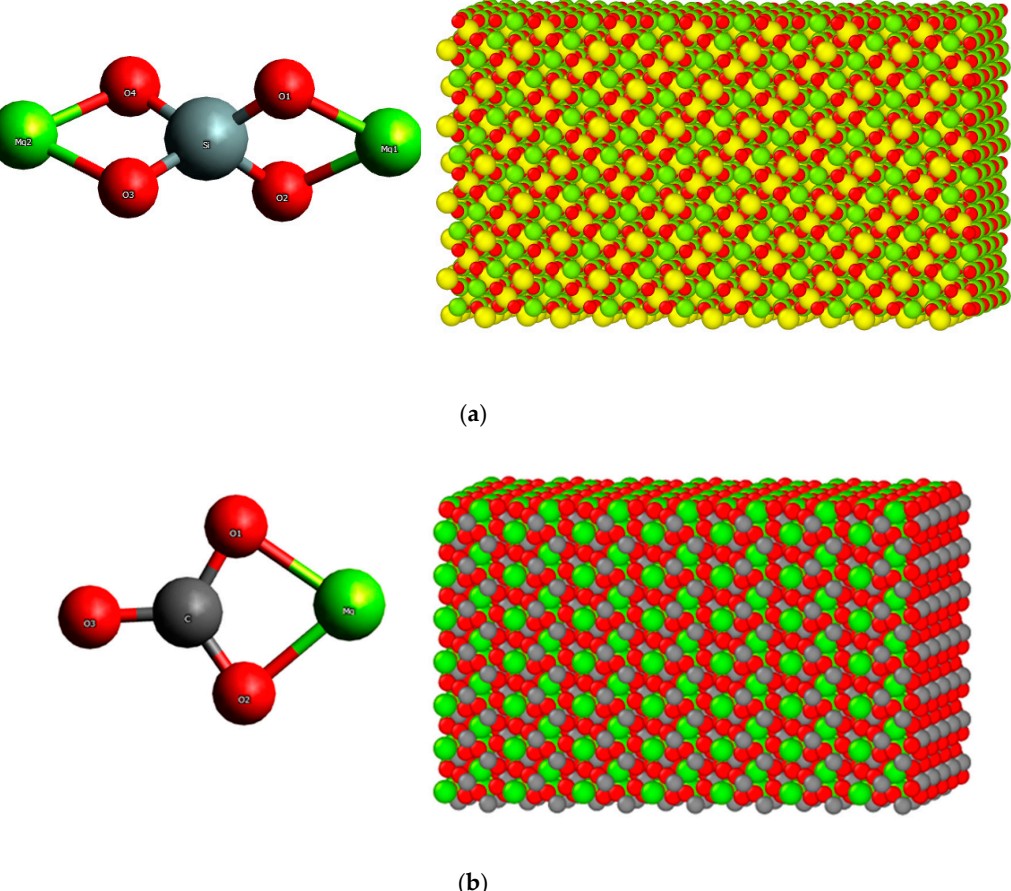

(**a**)

(**b**)

**Figure 1.** 3D atomic and crystal structure of forsterite (**a**) and magnesite (**b**).

The mineral carbonation reaction between forsterite and $CO_2$ is expressed as follows:

$$Mg_2SiO_4 + CO_2 \rightarrow MgCO_3 + SiO_2 \tag{1}$$

As mentioned before, mineral carbonation occurs in naturally occurring silicate rocks (alkaline/alkaline earth minerals) when a high concentration of $CO_2$ is brought into contact with them. This carbonation process can happen through both in situ and ex situ carbonation, which differ based on the time scale and contact of $CO_2$ [10]. In situ carbonation involves the injection of $CO_2$ into the geological formation containing silicate minerals (forsterite) to form solid carbonate minerals (magnesite) [11,12], while ex situ carbonation involves reacting silicate minerals with $CO_2$ in a controlled environment above ground. If the silicate minerals are brought from industrial wastes in the form of fly ash, cement kiln dust, steel slag, etc., they can sequestrate around 200–300 Mt of $CO_2$ annually [3,13,14]. This process sequestrates a large amount of $CO_2$ and produces reusable, valuable, and stable carbonate minerals for different industrial purposes [10]. However, both in situ and ex situ mineral carbonation processes are part of carbon-negative solutions and have a great potential to reduce the effects of $CO_2$ on the environment (Figure 2).

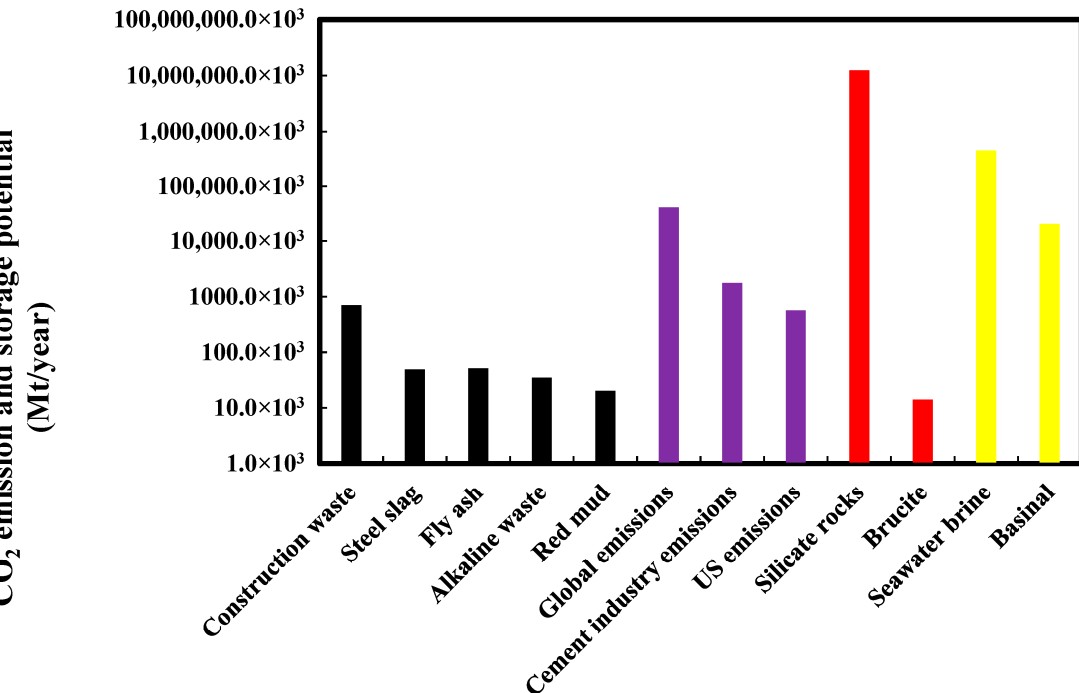

**Figure 2.** Global capacity to sequester $CO_2$ via mineralization [15].

The comminution process is a physical pre-treatment method in mineral processing that involves reducing the size of ore particles through crushing, grinding, and breaking the minerals into smaller/finer particles. It is a critical component for separating valuable minerals from waste rocks and requires significant energy input [16]. During the comminution process, factors such as the formation of minerals, mineral structure, presence of defects, mechanical strength, and grain size play a great role. The comminution process requires significant energy input to break rocks during crushing and grinding. $CO_2$ mineralization can reduce the comminution energy, as silicate-rich minerals require more energy for size reduction than carbonate [17]. Mineral carbonation can also change the surface chemistry of minerals, affecting the efficiency of both froth flotation and separation processes [18]. However, this study does not focus on explaining the reactive phenomena in the flotation process from a chemical point of view.

Comminution, froth flotation, and other mineral processes are affected by changes in the hardness and mechanical strength properties of minerals, which depend on crystalline structure, 3D arrangement of atoms, and interatomic behavior. Mineral carbonation alters these crystalline and interatomic structures. Few studies have been performed on the strength properties of forsterite and magnesite individually. Holyoke et al. [19] performed experiments on two different types of magnesite aggregation (coarse and fine grains) to determine the triaxial deformation over a wide range of temperatures (400–1000 °C) and strain rates ($2 \times 10^{-7}$ s$^{-1}$ to $2 \times 10^{-4}$ s$^{-1}$). In both aggregation types, the strengths of the magnesites at higher temperatures were reduced. Both fine- and coarse-grained magnesites showed little chance of recovery at the plastic stage. The study also mentioned that magnesite is more stable at low temperatures [19]. In another study, Liu et al. [20] carried out an atomic simulation (using a transferable empirical interatomic potential) to investigate the structural and elastic properties of magnesite over a wide range of pressure (based on the Earth's mantle's conditions). The simulation work found that magnesite shows anisotropic behavior at lower mantle depths but shows significant change with increasing depth. The percentage anisotropy in the shear and compressibility were calculated for a pressure range from 0 to 150 GPa and it was observed that, at higher pressure ($\geq$120 Gpa), both shear and compressibility values were close to 1 (0 means isotropic and 1 means anisotropic). This result means that magnesite is less stable at higher temperatures [20]. Yao et al. [21] worked on the impacts of pressure and temperature on magnesite using local-density approximation. All of the pressure and temperature values were in lower-mantle conditions. They found that the elastic and thermodynamic properties of magnesite were influenced by zero-point motion (the motion or vibration of the atoms at absolute zero temperature) and increasing temperatures. They also noticed a change (around 4.0%) in the shear and bulk modulus from static to ambient conditions (300 K and 0 Gpa). The authors considered 0 Gpa to indicate that the experiment was performed for Earth surface conditions, without any external pressure [21]. Gonzalez et al. [22] investigated the structural, dielectric, and vibrational spectroscopic properties of the amorphous form of forsterite. The work was conducted using two approaches: classical molecular dynamics (MD) for structural evolution using the empirical charge-based rigid ionic model and density functional theory (DFT) for measuring electronic structure using quantum mechanics. The radial distribution function (RDF) calculations for Mg–O and Si–O show broader profiles in increasing temperatures, which indicates the loss of crystallinity of forsterite. This work also analyzed the degrees of freedom for disordering in the crystal structure at higher temperatures (400 K to 2000 K). The increasing temperatures accelerated the dynamics of melting of forsterite and at some point (above 1800 K), the system became unstable for the disorganization of the interatomic positions [22]. In 2019, Gouriet et al. investigated the mechanical deformation of the orthorhombic structure of forsterite under applied strains. They determined the energy–strain curves and linear elastic regimes to ascertain the ultimate instability of the crystal structure. The maximum stress values from stress–strain curves were found to be approximately 15.9, 12.1, and 29.3 Gpa along the different [001], [010], and [100] directions, respectively. They also observed a similar change in features in ideal shear stress evolution in each direction. The shear stress increased initially, then decreased for a percentage of strain (around 10%) due to the bond divergence between Mg–O. But with the reduction of bond distance between Mg–O, the shear stress as well as stiffness increased again with increasing strain [23]. In addition, Choudhary et al. (2020) studied the mechanical stability and degradation of forsterite and noted that increasing content of forsterite increased the mechanical strength of a composite compared to calcium phosphates and calcium silicates [24]. Several studies also worked on generalizing the relationship between the compressive strength of materials and the grain size of the materials and stated that there is a growing trend of mechanical strength with increasing grain size [25,26]. These studies investigated the petrophysical and mechanical properties of carbonate minerals for different grain sizes and claimed that decreasing the average grain size reduces the strength properties of materials.

Until now, no studies have investigated and compared the mechanical properties of forsterite and magnesite in various conditions (temperature, pressure, and applied forces). This present work focuses on the evolution of the stress–strain relationship according to changes in physical properties. The effects of temperatures, loading rates, and grain sizes on the maximum stress values of two different polycrystals were studied. In addition, polycrystals highly sensitive to the applied forces are also measured. Changes in pressure and thermal effects induced on the elastic properties are observed for the two polycrystals. Later, the radial distribution function is also evaluated for measuring the pairwise interaction between Si–O and C–O bonds. This study compares the strength properties and microstructure of the two polycrystals under the same conditions, to assess the effect of mineral carbonation on comminution energy for mineral processing.

## 2. Molecular Modeling and Simulation Method

Molecular dynamics (MD) simulation provides a basic representation and interpretation of molecular interaction modeling of any material with detailed information and underlying governing mechanisms. It is a powerful tool for studying particle movement, mechanical behavior, and other properties according to the materials' physical characteristics and chemical reaction kinetics [27]. Using MD, deformation, elasticity, diffusion, yield, and other physical behaviors of materials at the atomic level can be measured. MD can predict the dynamics of materials' behavior under different temperatures and loading conditions by performing computational experiments. The benefits of using MD are flexibility in working with different sizes of materials and visualization of the molecular interaction at the atomic scale which might not be possible in experiments. The fundamental theory of molecular dynamics simulation is to observe the dynamic trajectory of an atomic system and analyze the atomic interactions among the respective atoms by solving Newton's equation of motion. Prior works on different materials using molecular dynamics simulation have offered reliable outputs for illustrating their mechanical properties.

Here in this paper, a molecular dynamics simulation study on forsterite and magnesite polycrystals is performed using the Large-scale Atomic/Molecular Massively Parallel Simulator (LAMMPS) (9 October 2020) software package [28].

Better performance of molecular dynamics simulation depends on the implementation of a suitable and successive potential for polycrystals. Since the model's reliability and validation directly depend on this selected potential, using the right forcefield parameters to obtain accurate results and configuration is important. The accessibility of the interatomic potential to the size of the system is a governing factor for the numerical integration of the above model. Here, in this study, an empirical potential model named the 'Buckingham' potential is used for simulating the atoms of both polycrystals [29]. This potential is freely available with the LAMMPS library package. It considers long-range electrostatic terms with classical Coulombic energy, a short-range repulsive term, and a three-body harmonic term. This potential is a bond-order and short-range interaction-based potential which is attributed to the partial charge of the atoms. Several prior studies have used this potential to reproduce the structural properties of both forsterite and magnesite polycrystals [30,31]. The Buckingham potential can be expressed as follows:

$$E = A_{ij}.e^{-\frac{r_{ij}}{\rho_{ij}}} - \frac{C_{ij}}{r_{ij}^6} \tag{2}$$

$$E = \frac{Cq_iq_j}{\epsilon r_{ij}} \tag{3}$$

where $E$ is the potential energy, $A$ is energy units, $e$ is the elementary charge, $\rho_{ij}$ is the hardness parameter of repulsive energy (indicates the characteristic length scale that determines the range over which the potential decays exponentially), $C$ is the coefficient of dispersive energy, $r_{ij}$ is the interatomic distance between two atoms $i$ and $j$ ($i$ and $j$ being Mg, Si, C, and O), and $q_i$ and $q_j$ are the atoms' partial charges. Though the Buckingham

potential seems simple, this potential considers the pair's ion charges, thermal expansion, heat capacity, and long-range coulombic interaction to illustrate the short-range interaction between the paired atoms. For this reason, this potential works slowly compared to other interatomic potentials used mainly for silicate and carbonate molecules. The performance of this potential depends on some forcefield parameters of the forsterite and magnesite. The crystal structures for forsterite and magnesite are considered orthorhombic and trigonal, respectively, and the crystal orientations are made with specified lattice constants. The volume of the simulation models for forsterite and magnesite are 125 nm$^3$ and 117.65 nm$^3$. The initial configuration and forcefield parameters are shown in Tables 2 and 3, respectively.

**Table 2.** Selected materials and atomic model properties.

| Material Type | Dimension (Å) | Potential | Number of Atoms | Atomic Bond Type | Lattice Constant (Å) | Crystal System |
|---|---|---|---|---|---|---|
| Forsterite ($Mg_2SiO_4$) | $50 \times 50 \times 50$ | Buckingham | 4480 | Ionic–Covalent | a = 4.787<br>b = 10.272<br>c = 6.023 | Orthorhombic |
| Magnesite ($MgCO_3$) | $49 \times 49 \times 49$ | Buckingham | 4250 | Ionic–Covalent | a = 4.64<br>b = 4.64<br>c = 14.93 | Trigonal |

**Table 3.** Forcefield parameters used in this study.

| **Buckingham Potential** | | | |
|---|---|---|---|
| Ion Pairs | A(eV) | B(Å) | C(eVÅ$^6$) |
| Mg-O | 1428.5 | 0.2945 | 0 |
| Si-O | 473.2 | 0.4157 | 0 |
| O-O | 22,764.30 | 0.149 | 60.08 |
| C-O (Morse) | 4.71 | 3.8 | 1.18 |
| **Harmonic 3-Body Term** | | | |
| | k(eV rad$^{-2}$) | $\Theta_0$ (degrees) | |
| O-Si-O | 2.09 | 109.47 | |
| O-C-O | 1.69 | 120 | |
| **Charges** | | | |
| Mg | +2.00 | | |
| Si | +4.00 | | |
| C | +1.135 | | |
| O (for forsterite) | +0.84819 | | |
| O (for magnesite) | −1.632 | | |

In this study, Moltemplate was used to generate all-atom molecular models for forsterite and magnesite [32]. Moltemplate is a cross-platform text-based molecular builder (for both all-atom and coarse-grained molecular models) made for LAMMPS. The simulation models of these two polycrystals were made with random orientations of the polycrystals as well as by using the Voronoi method including random seeds. The conjugate gradient (CG) algorithm was used to optimize the initial structure and position of the atoms. This minimization algorithm adds the force gradient to the previous iteration's information to compute the new direction perpendicular to the previous search iteration. The Verlet algorithm was used with a timestep of 1 *fs* to calculate the dynamic trajectories of the particles. The long-range Coulombic interaction was calculated using the Ewald summation method. Two different systems approached a specified temperature (300 K) using an NPT ensemble under zero pressure. A Nose–Hoover thermostat and barostat were used successfully to control the temperature and pressure. Periodic boundary conditions were applied in each direction of the simulation box. The tensile force was applied to the polycrystals by using incremental homogeneous strain, which indicates the displacement of the atomic layers along the tensile direction. The components of stress tensors were

brought to zero at each deformation state so that the atoms received enough time for relaxation. Before the measurement of the stress–strain properties, in all cases, the polycrystal structures were optimized for equilibrium purposes.

## 3. Results and Discussion

### *Stress–strain behavior of the forsterite and magnesite*

Multiple multiscale approaches have been developed to determine the strength properties of materials, but the determination of the stress–strain relationship for different thermodynamic conditions is one of the most reliable options among relevant approaches. These material properties should be determined by allowing deformation in the designed microscale models under uniaxial tensile or compression tests as these properties are considerably changed by different strain rates and temperatures. The definition and idea of continuum Cauchy stress in atomistic simulation are slightly different but equivalent to the definition of Virial stress [33]. For stress calculation, the per-atom pressure tensor is computed for each atom in the group.

Here, both uniaxial tensile and compression tests are performed to study and compare the stress–strain behavior of these two polycrystals for different strain rates and temperatures. Three different temperatures (300 K, 500 K, and 700 K) for a constant strain rate of 0.01 ps$^{-1}$ and three different strain rates (0.01 ps$^{-1}$, 0.03 ps$^{-1}$, and 0.05 ps$^{-1}$) for a constant temperature at 300 K are considered during deformation under uniaxial tensile and compression tests. For both polycrystals, the evolution of the stress values as a function of strain values is studied along the [100] plane in the x direction.

### *3.1. Stress–Strain Behavior of the Geomaterials for Different Temperatures*

Figure 3a shows a typical example of a Young's modulus-based stress–strain curve used to study the mechanical properties of the two geomaterials. The curve depicts that the ultimate strength/stress point is the maximum stress for any material before failure occurs under tensile or compression load. The elongation of the elasticity of materials depends on the values of this maximum stress point. Before reaching this point, the loading stress increases with increasing strain, but after the peak point, the material faces deformation. In this region, the stress values usually decrease with increasing strain values. Figure 4a,b represents such types of stress–strain relationships of the studied polycrystals (forsterite and magnesite, respectively) under a constant tensile load for temperatures 300 K, 500 K, and 700 K, at a strain rate of 0.01 ps$^{-1}$. As expected, a parabolic evolution of the stress values corresponding to the initial linear portion of the stress–strain curves is observed for both polycrystals. For forsterite (Figure 4a), the ultimate strength point is near 26 GPa at 300 K, and the Young's modulus value is around 154.7451 GPa, and this value was validated using the result of the Young's modulus (153.2 GPa) calculated by Gouriet et al. (shown in Figure 3b) [23]. Again, at 500 K and 700 K, the corresponding ultimate strength values are found to be near 21.79 GPa and 17.92 GPa, respectively. The increasing temperatures from 300 K to 500 K and 700 K initiated considerable drops in the maximum stress of forsterite by 16.15% and 31.07%. More importantly, increasing temperature also increased the maximum strain values needed to obtain the maximum stress point. For instance, at 300 K, the maximum stress point was achieved for a strain value of 0.168, whereas at 500 K and 700 K, maximum stress points were obtained at the strain values of 0.181 and 0.199, respectively.

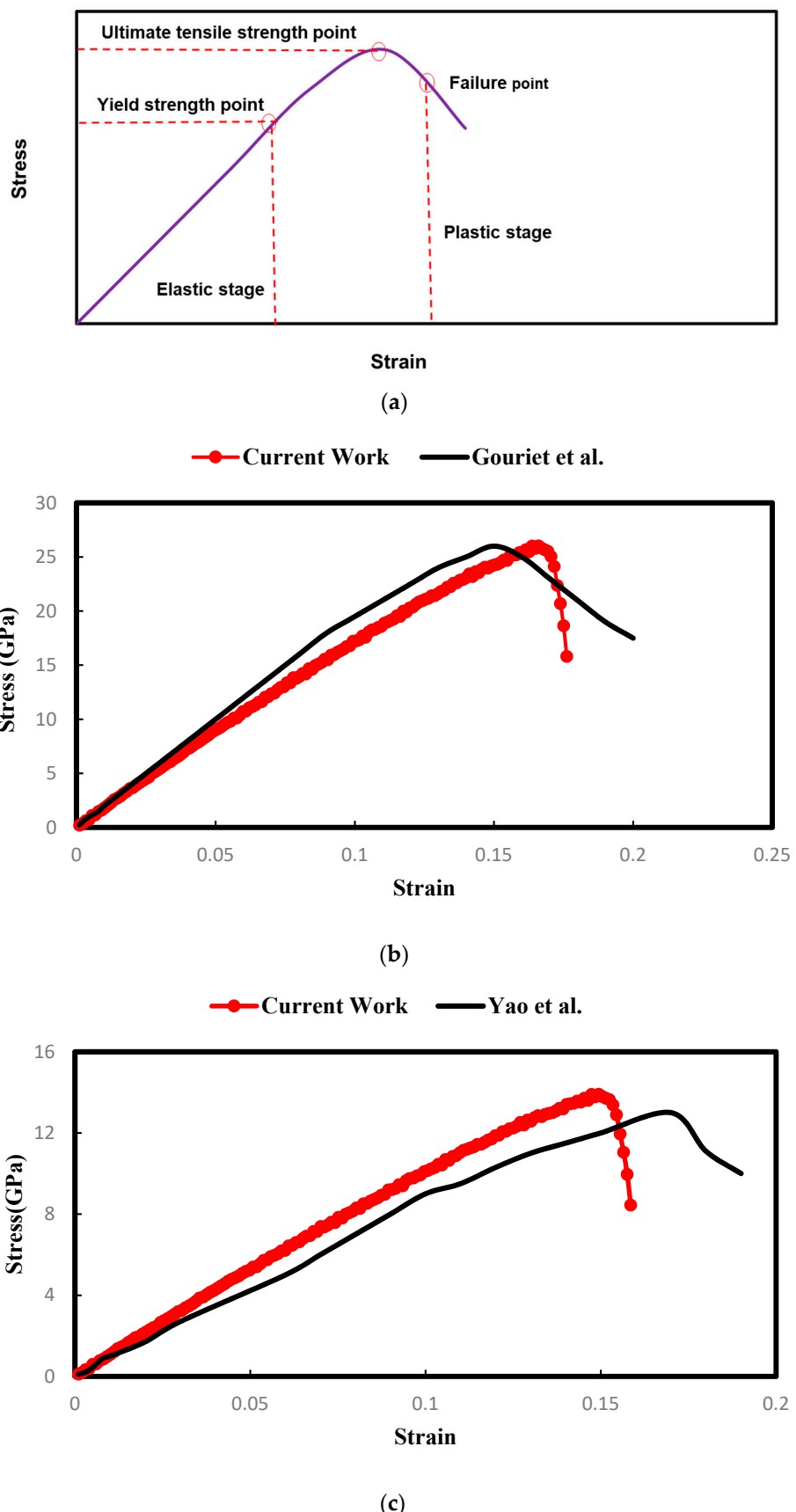

**Figure 3.** (**a**) A generalized physical model of the stress–strain relationship for any material. (**b**) The validation of the work for forsterite at 300 K [23]. (**c**) The validation of the work for magnesite at 300 K [21].

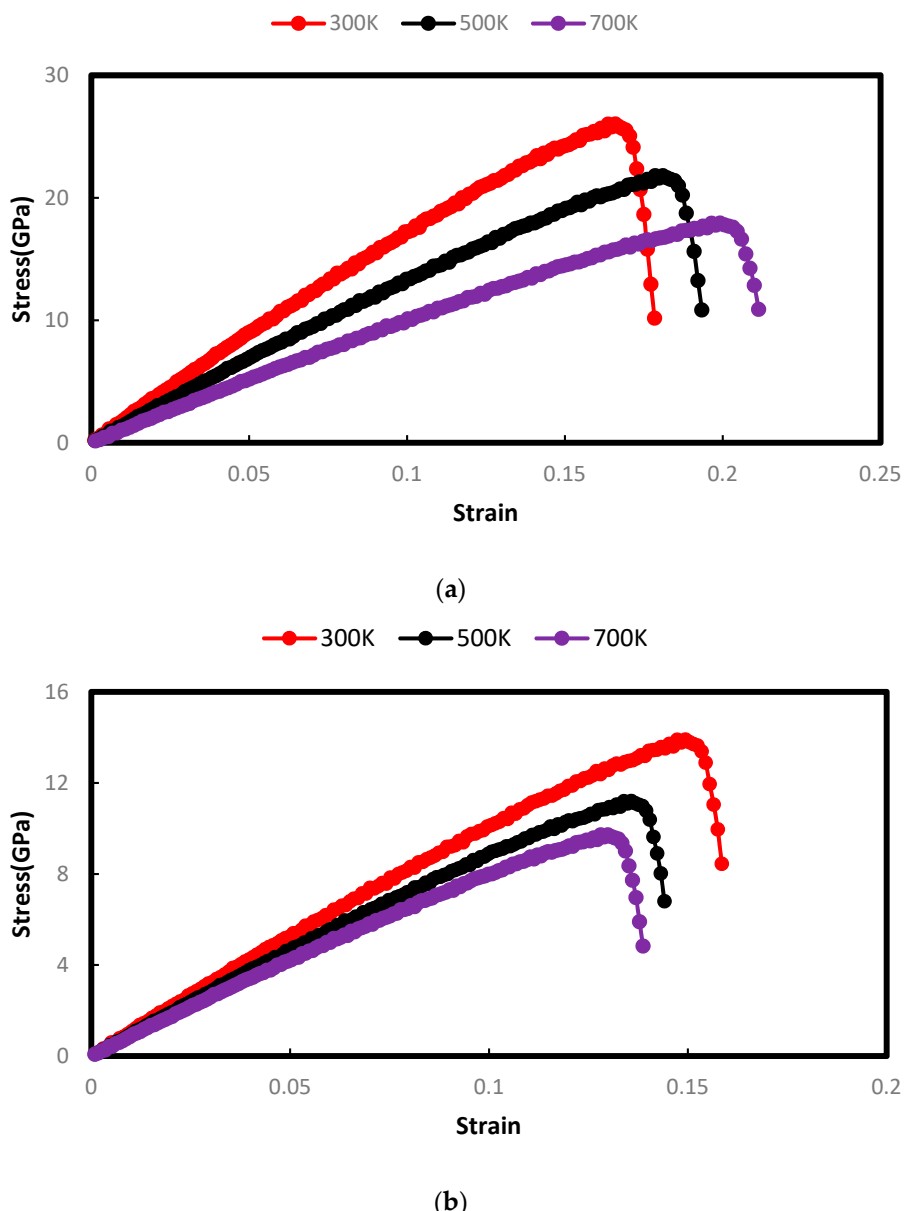

**Figure 4.** (**a**) The stress–strain curve of forsterite for different temperatures under tensile test (at a strain rate of 0.01 ps$^{-1}$). (**b**) The stress–strain curve of magnesite for different temperatures under tensile test (at a strain rate of 0.01 ps$^{-1}$).

From Figure 3c, the ultimate strength point for magnesite is found near 13.90 GPa (at 300 K) with a corresponding value of Young's modulus of 92.84 GPa. This value was validated using the calculated result (ultimate strength of 13.70 GPa at 300 K) of Yao et al. (shown in Figure 3c) [21]. Then, at 500 K and 700 K, the ultimate strength values are decreased by 19.42% and 34.46%, respectively. In addition to that, as opposed to those of forsterite, the maximum strain values for magnesite are decreased by 10.06% (for 500 K) and 13.42% (for 700 K) as the temperature is changed from 300 K. This behavior indicates that at elevated temperatures, forsterite tends to show more ductility than magnesite. Table 4 provides information on the Young's modulus values of forsterite and magnesite for different temperatures.

**Table 4.** Ultimate strength and Young's modulus of forsterite and magnesite under tensile force.

| | Ultimate Strength (Gpa) | | | Young's Modulus (Gpa) | | |
|---|---|---|---|---|---|---|
| | 300 K | 500 K | 700 K | 300 K | 500 K | 700 K |
| Forsterite | 26.00 Gpa | 21.80 Gpa | 17.92 Gpa | 154.74 | 120.79 | 89.98 |
| Magnesite | 13.90 Gpa | 11.20 Gpa | 9.12 Gpa | 92.84 | 83.58 | 70.67 |

Figure 5a,b shows the stress–strain curves of forsterite and magnesite under the uniaxial compression test, respectively, under the same conditions as for the uniaxial tensile test. The stress–strain relations under this compressive force represent a trend similar to that under tensile force at different temperatures. Since compressive force usually provides more strain energy to materials, the ultimate stress loading stage occurs earlier than in the case of tensile force. This increasing strain energy results in the material entering the plastic stage. For forsterite (Figure 5a), the ultimate stress point is observed at 17.28 GPa, at 300 K, and at a strain value of 0.143; the calculated Young's modulus is about 120.457 GPa. Increasing the temperature to 500 K and 700 K reduces the maximum stress point to approximately 15.12 GPa and 13.45 GPa, respectively. These results indicate that the effect of temperature on stress–strain properties under compressive forces is similar to those for tensile forces. The only difference is that, due to the contraction under compressive forces applied to the crystal material, the elastic stage ends sooner compared to the elongation for tensile forces. For example, under compressive stress at 300 K, the elastic stage comes to an end at a maximum stress point of 17.28 GPa (Figure 5a), whereas, for tensile stress, it reaches 26 GPa (Figure 4b). In magnesite (Figure 5b), the increasing stress loads (up to a strain value of 0.107) cannot provide any significant change in the elastic stage under compressive force for each temperature studied (300 K, 500 K, and 700 K). However, the maximum values of the stress load of the magnesite are lower for increasing temperature. This behavior of the crystals under tensile and compressive forces has indicated that increasing temperature (300 K) results in deformation for both crystals as high temperatures contribute to initializing local stress for propagation/stretching out of the crystals. However, the results also show that increasing the temperature from 300 K to 500 K and 700 K under compressive force reduces the Young's modulus of forsterite (from 120.457 GPa to 79.98 GPa, respectively) more than magnesite (from 77.04 GPa to 60.88 GPa, respectively).

The effect of increasing temperature is the same for both polycrystals under two different loading velocities. In both cases, increasing temperature leads to smaller peaks of stress, but with higher ductility (for forsterite) and lower ductility (for magnesite) up to the failure point of the polycrystals. Forsterite shows more ductility at elevated temperatures than magnesite. The results obtained from the uniaxial tensile and compressive force tests for the two polycrystals at a fixed temperature (600 K) are shown in Figure 6 Table 5. In addition, Figure 7a,b shows the evolution of strain energy as a function of strain for a constant temperature (at 600 K) to understand the change in energy during the application of force. As seen in Figure 7b, the strain energy of the microstructure (after the relaxation period) changes with strain values. For both polycrystals, each curve has an inflection point indicating the maximum strain energy the minerals can tolerate. This peak represents the polycrystals' maximum stress point (ultimate strength). After crossing the peak, the system starts dissipating energy which results in the deformation of the polycrystals. This dissipation of energy moves the strain energy curve in the downward direction for additional strain values. Similar findings have been noted in the strain energy curve. Forsterite shows higher ductility and elastic properties for both types of applied forces (strain energy ranges from 12.02–15.05 meV/$A^3$) compared to magnesite.

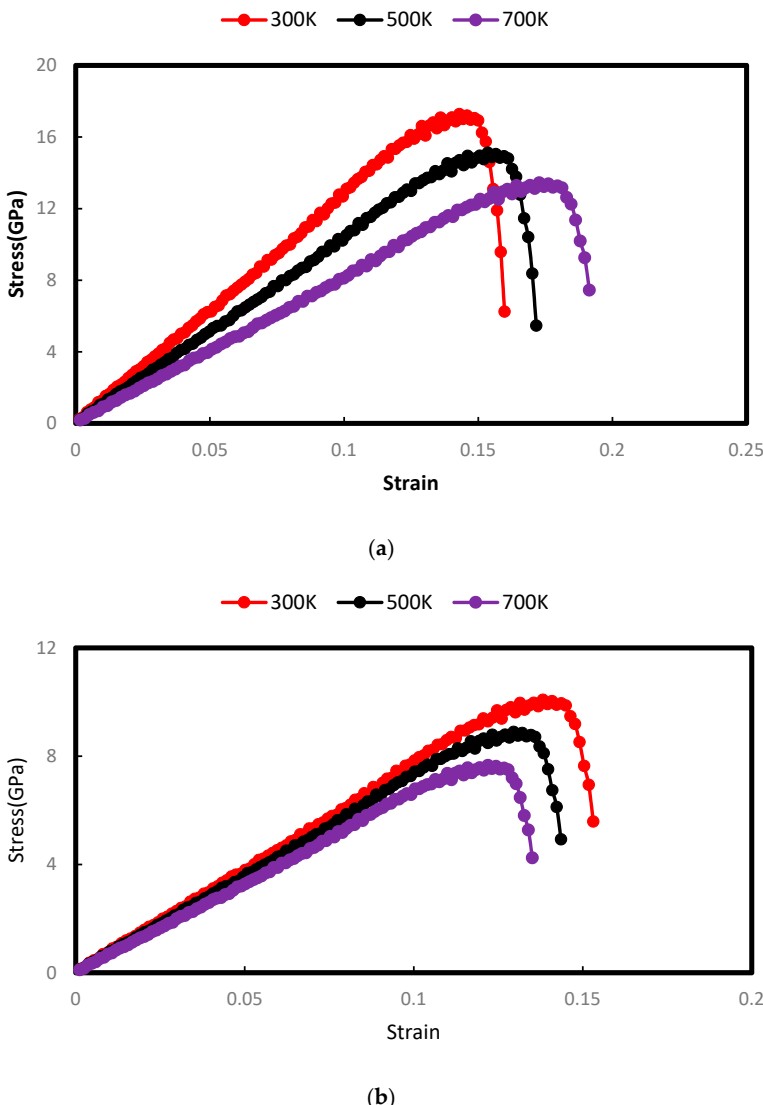

(**a**)

(**b**)

**Figure 5.** (**a**) The stress–strain curve of forsterite for different temperatures under compression test (at a strain rate of 0.01 ps$^{-1}$). (**b**) The stress–strain curve of magnesite for different temperatures under compression test (at a strain rate of 0.01 ps$^{-1}$).

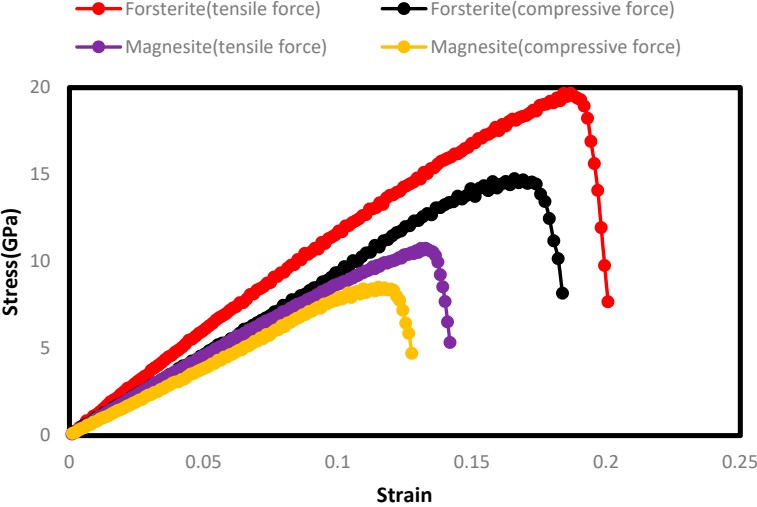

**Figure 6.** The stress–strain curves for a constant temperature (600 K) under different test conditions.

**Table 5.** Comparison of the obtained results at 600 K for two different applied forces.

| Material Type | Temperature | Maximum Stress Reduction from Tensile Force to Compressive Force (%) | Strain Variation in Achieving Maximum Stress Loading Capacity for Different Applied Forces (%) |
|---|---|---|---|
| Forsterite | 600 K | 25.01 | 11.18 |
| Magnesite | 600 K | 20.12 | 13.26 |

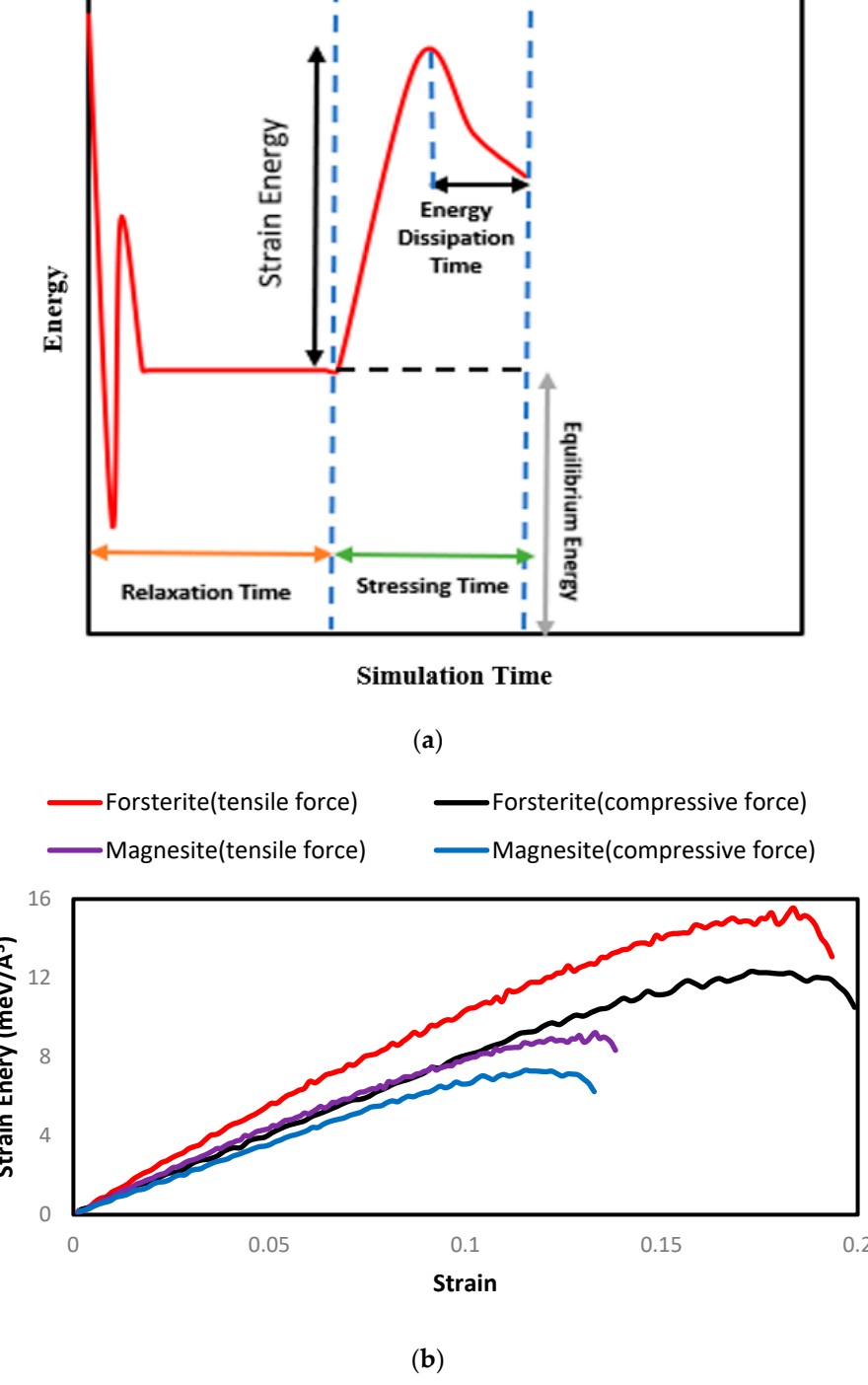

**Figure 7.** Evolution of strain energy as a function of strain: (**a**) a generalized curve, (**b**) for both forsterite and magnesite at 600 K.

### 3.2. Stress–Strain Behavior of the Geomaterials for Different Strain Rates

In this section, different strain rates are considered to study the impact of loading velocity on the mechanical deformation of both polycrystals. For this purpose, three different loading rates (0.001, 0.01, and 0.05 ps$^{-1}$) are applied under both tensile and compressive forces. The effect of different loading rates has been studied previously on different crystals using molecular dynamics simulation. It should be noted that the polycrystals were in a relaxation state before applying both tensile and compressive forces. Therefore, the stress values as a function of strain values start from the undeformed state of the polycrystal (at a strain value of zero) either for increasing or decreasing loading velocities. Figure 8a,b shows the stress–strain curves for forsterite and magnesite at a constant temperature of 300 K (under tensile force), respectively, for the three abovementioned loading rates. According to the results demonstrated in the figures, increasing strain rate increases the maximum stress points for both polycrystals. Particularly, the elastic stage is increased for increasing strain rates with negligible impact on the plastic stage. The yield points are only changed for different loading velocities. In addition, the strength of non-viscoelastic materials like crystals greatly depends on the loading rate; a higher strain rate indicates a higher strength of the material. For forsterite (Figure 8a), increasing the strain rate from 0.01 ps$^{-1}$ to 0.05 ps$^{-1}$ increases the maximum stress point by 10.71%, and decreasing the strain rate to 0.001 ps$^{-1}$ decreases it by 11.43%. For magnesite (Figure 8b), the maximum stress point moves to 15.21 Gpa for the loading rate of 0.05 ps$^{-1}$, whereas the peak stress point is decreased by 18.30% to 0.001 ps$^{-1}$ when the strain rate decreases. Under application of compressive force, the changes in the stress–strain relationship follow a similar trend for both materials like the tensile force. Figure 9a shows the results for forsterite, whereas the changes for magnesite are shown in Figure 9b.

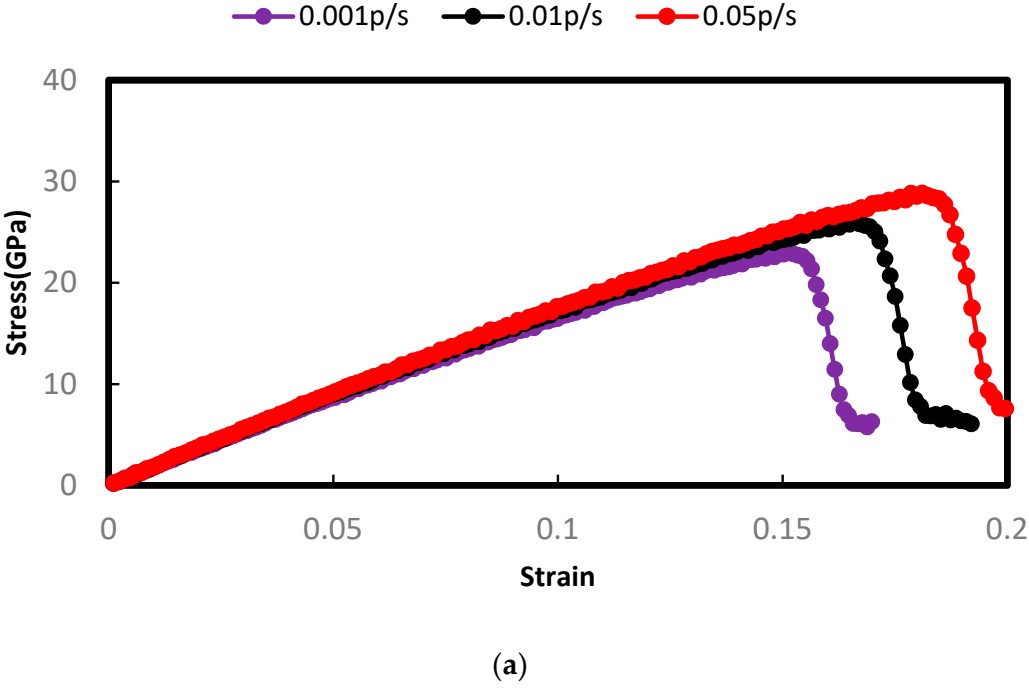

(**a**)

**Figure 8.** *Cont*.

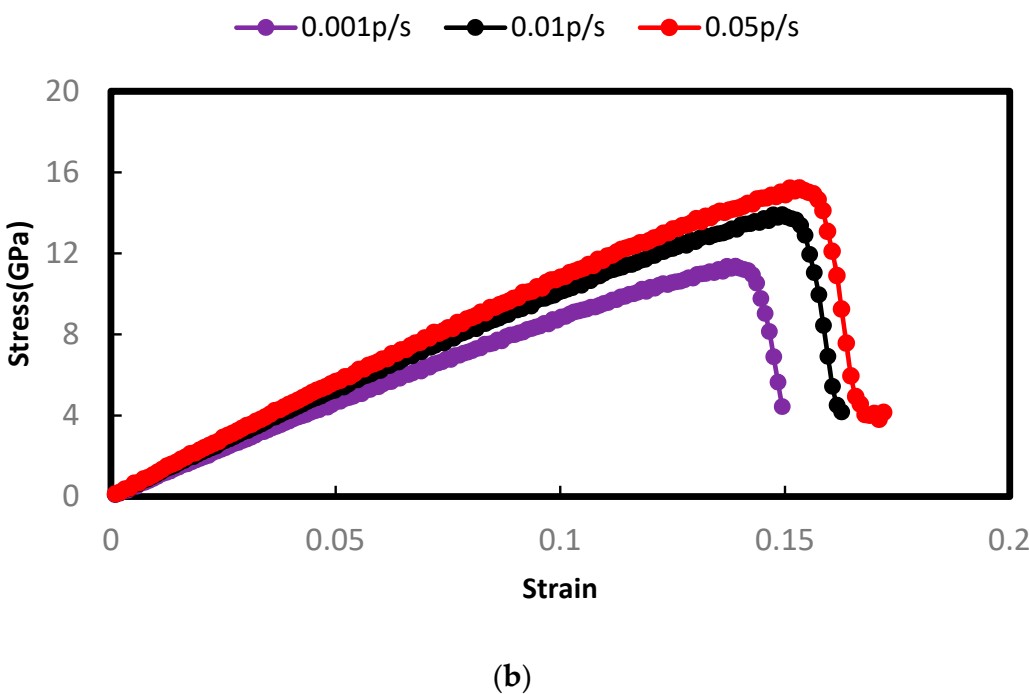

(**b**)

**Figure 8.** (**a**) The stress–strain curve of forsterite for different strain rates under tensile test (at 300 K).
(**b**) The stress–strain curve of magnesite for different strain rates under tensile test (at 300 K).

Here, for higher strain rates, the values of strain energy increase for both forsterite
and magnesite to initiate plastic deformation. This is contradictory to the case of higher
temperatures. The higher strain rate provides a smoother trend of the stress–strain rela-
tionship (more linear) which leads to shortening the breaking time during deformation.
This higher strain rate makes the elastic region more linear, which is also described by
Hooke's law [34]. This required energy is higher for tensile forces than for compressive
forces. Hence, forsterite shows more strength at a higher strain rate compared to magnesite.
The results of the stress–strain relationship between these two polycrystals obtained under
both types of forces are compared for a constant strain rate (at 300 K and a strain rate of
0.03 K) and are shown in Figure 10.

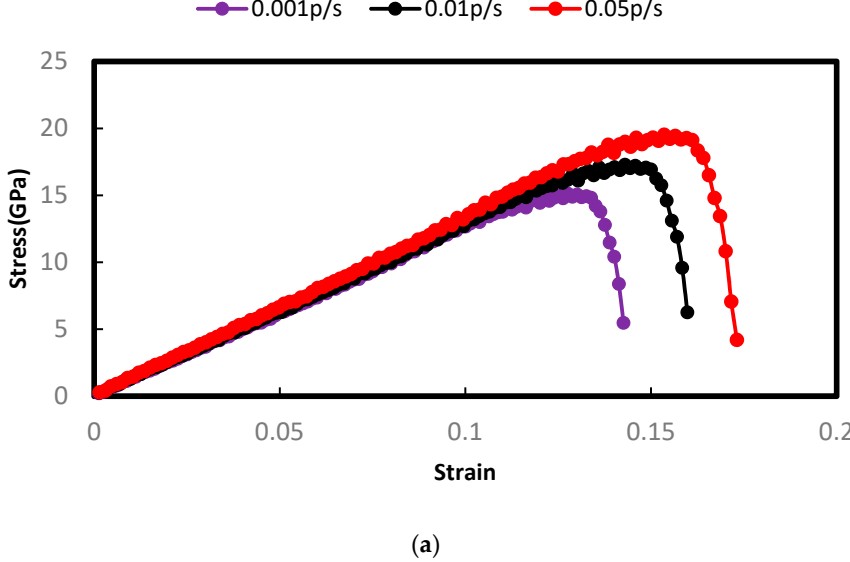

(**a**)

**Figure 9.** *Cont.*

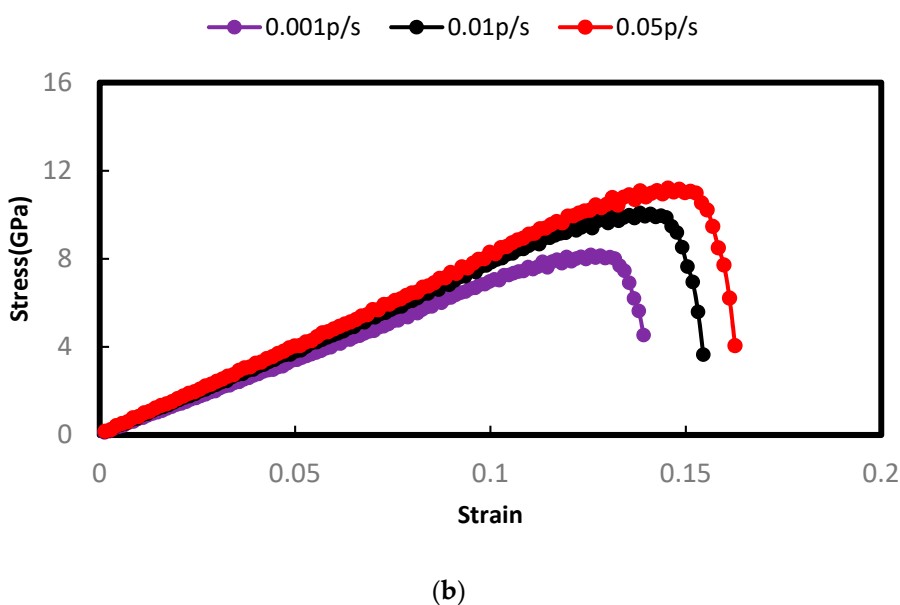

(**b**)

**Figure 9.** (**a**) The stress–strain curve of forsterite for different strain rates under compression test (at 300 K). (**b**) The stress–strain curve of magnesite for different strain rates under compression test (at 300 K).

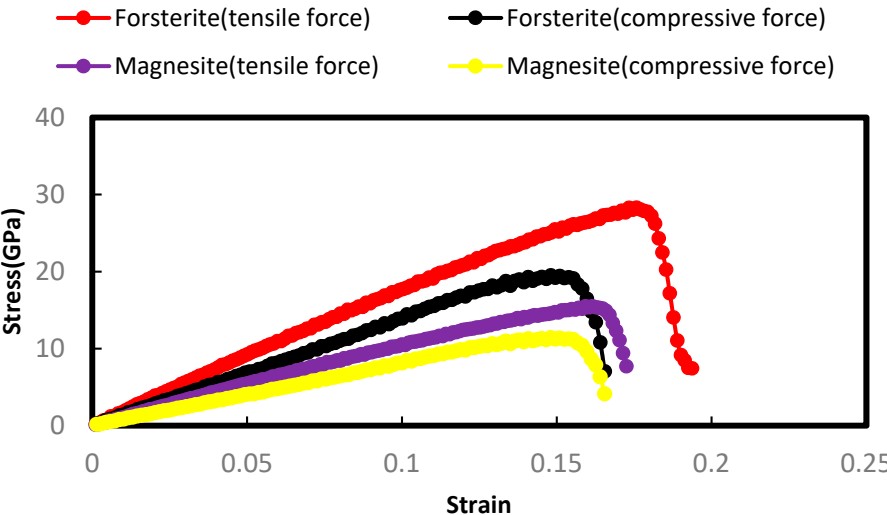

**Figure 10.** The stress–strain curves for a constant strain rate under different tests.

### 3.3. Strain Rate Sensitivity

The strain rate sensitivity (SRS) measures the change in yield pressure for different strain rates. Sometimes, applied force alters the characteristics of the material which can be understood by SRS calculation. This is a relationship wherein a material's tensile strength depends on different loading rates. Based on the conditions of a steady-state process, the SRS of a material in loaded tension can be demonstrated (as a differential form) by Hart [35] as follows:

$$m = \frac{\partial \log(\sigma)}{\partial \log(\dot{\varepsilon})} \tag{4}$$

where *m* is the strain rate sensitivity, σ is the tensile strength, and $\dot{\varepsilon}$ is the strain rate.

Hence, Equation (4) states that the SRS values can be obtained from the slopes of the logarithmic plot of tensile stresses and corresponding strain rates. Hart [35] exhibited that stress rates are proportional to strain rates up to the ultimate/maximum strength

point, and during plastic deformation, the material is less sensitive to the extra load. This sensitivity impacts the material's stability during deformation due to strain localization. The SRS shows the ability of a material to resist necking and maintain its stability during deformation. However, higher temperatures reduce the SRS of materials, and the higher the values of SRS are, the higher the strength of materials is. For higher SRS, materials are able to distribute force more evenly through their structure, rather than localizing it in one area.

Here, simulations were performed for three different temperatures (300 K, 500 K, and 700 K) and three different strain rates (0.001, 0.01, and 0.05 ps$^{-1}$) to determine and compare the obtained SRS results for forsterite and magnesite. Here, steady strain rate sensitivity is considered, where the strain rate sensitivity coefficient *m* is computed using the above-mentioned equation for different strain rates. The negative sign is neglected here in the obtained values of m. The stability of the material is proportional to the strain rate sensitivity. A higher value of m indicates a large change in the flow stress for a change in strain rate. Figure 11a,b shows the variation in *m* values with strain rates for forsterite and magnesite, respectively, for different temperatures and strain rates. For forsterite and magnesite (Figure 11a,b), at a constant strain rate (0.01 ps$^{-1}$), increasing the temperature from 300 K $\rightarrow$ 500 K $\rightarrow$ 700 decreases the *m* value from 0.70764 to 0.66992, followed by 0.62671 and from 0.5716 to 0.52972 and followed by 0.47712. Both forsterite and magnesite become less stable at higher temperatures. On the other hand, for a constant temperature (300 K), when the strain rates increase, the *m* values increase, for both forsterite and magnesite (known as strain rate hardening). Hence, both materials show high sensitivity to increasing strain and more stability for higher strain rates. In short, higher temperatures decrease the sensitivity of materials to strain rate and decrease the yield strength.

From the above results, it has been found that forsterite shows higher sensitivity than magnesite, which means that magnesite is comparatively less stable. The comparison of the results for both polycrystals is shown in Figure 12. At a constant temperature (500 K), the SRS values of forsterite and magnesite are 0.43128 and 0.32405, 0.66922 and 0.52972, and 1.0473 and 0.8618 for the strain rates of 0.001 ps$^{-1}$, 0.01 ps$^{-1}$ and 0.05 ps$^{-1}$, respectively. However, despite having a lower strain rate effect at lower temperatures, the effect of temperature on SRS is significantly stronger for lower strain rates as the sensitivity arises from the inertness of the defective structure evolution of materials for lower strain rates [36]. Since the strength of the materials is proportional to the SRS, both polycrystals are prompted to deform earlier when the SRS values are lower for corresponding temperatures and strain rates. The impact of temperature is found to be higher in tensile strength for magnesite than forsterite which agrees with the stress–strain curves in the previous section. In short, less sensitivity to applied forces makes magnesite weaker than forsterite.

### 3.4. Effect of Grain Size

The impact of grain size on both polycrystals is studied here. Polycrystals are comprised of multiple grains, and these grains are bounded by some interfacial defects in the grain boundaries of those polycrystals. These grains' structure and energy provide microscopic insight into the mechanical deformation of the polycrystals. Particularly, the morphology, size distribution, and nature of these grains and grain boundaries are important features of the polycrystals [37–39]. Also, the grain size and boundaries resist the elongation of the polycrystals and reduce the yield strength and ductility of the materials. However, there is a significant correlation between grain size and the mechanical strength properties of minerals (particularly in sedimentary rocks) which has been proven by previous studies [40]. Studies on grain size microstructure provide insights into materials' toughness, corrosion resistance, thermal conductivity, and magnetic susceptibility.

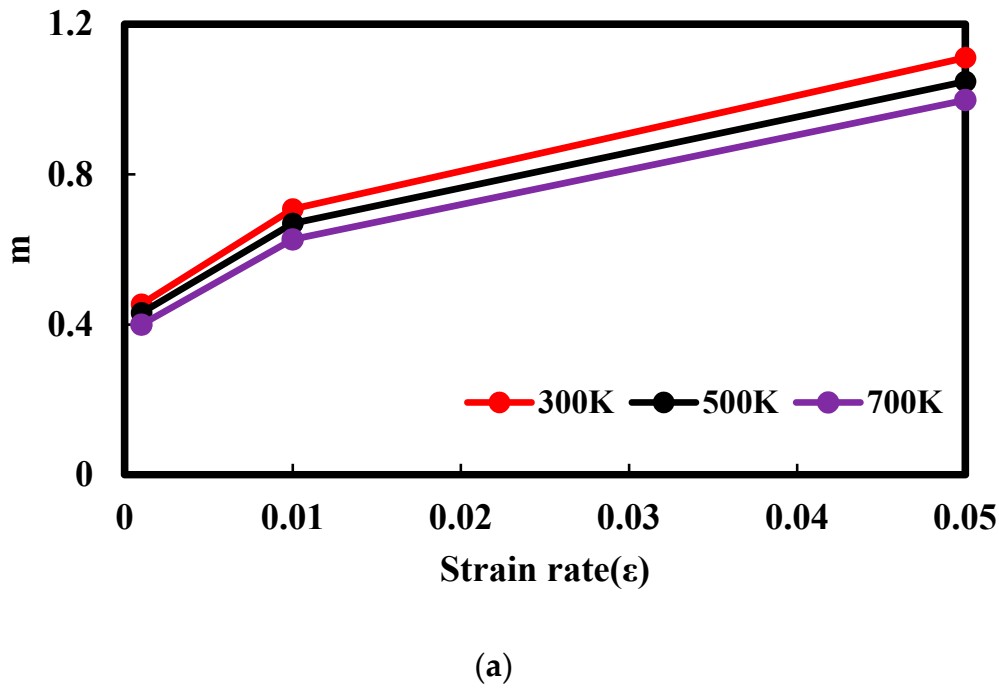

(**a**)

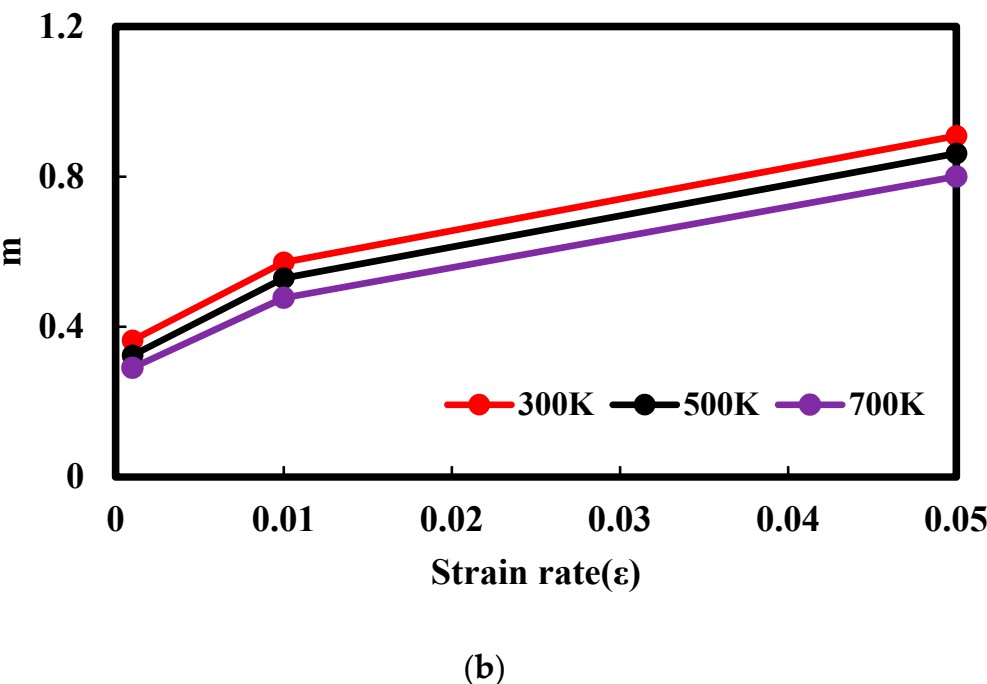

(**b**)

**Figure 11.** Strain rate sensitivity of (**a**) forsterite and (**b**) magnesite, for different temperatures (300 K, 500 K, and 700 K) and strain rates (0.001 ps$^{-1}$, 0.01 ps$^{-1}$, and 0.05 ps$^{-1}$).

Since there is a relationship (nonproportional) between grain size and density of the materials, in this study, the number of atoms is increased for both minerals to study the impact of grain size. This is because a greater number of atoms are more closely packed and result in higher densities of materials, manifesting as smaller grain sizes [41]. Here, the grain size is calculated by measuring the average distance from the two adjacent peaks in the radial distribution function for both studied polycrystals. Figure 13 shows the yield strengths of both forsterite and magnesite at a constant strain rate (0.01 ps$^{-1}$)

for 5600 atoms and 4850 atoms, respectively. This figure shows that the polycrystals are more deformative for growing numbers of atom. The yield strengths of the forsterite and magnesite dropped by 7.89% and 9.09% compared to the initial systems. These results show an opposite correlation between compressive strength and number of atoms. From the figure, it is observed that the grain size also impacts the Young's modulus values of the two polycrystals (dropped to 15.92 and 9.17 GPa, respectively). In the case of the two studied polycrystals, the only components bearing the applied stress are grains. In addition, the distribution of the grains becomes uniform for higher numbers of atoms, hence the deformation of the polycrystals occurring earlier under pristine conditions with a low number of atoms. Shear velocity, wave velocity, relatively low energy, and partial double-bond character in Si–O and C–O bonds also cause a reduction in strength properties for lower grain sizes; however, these discussions are not part of our current study.

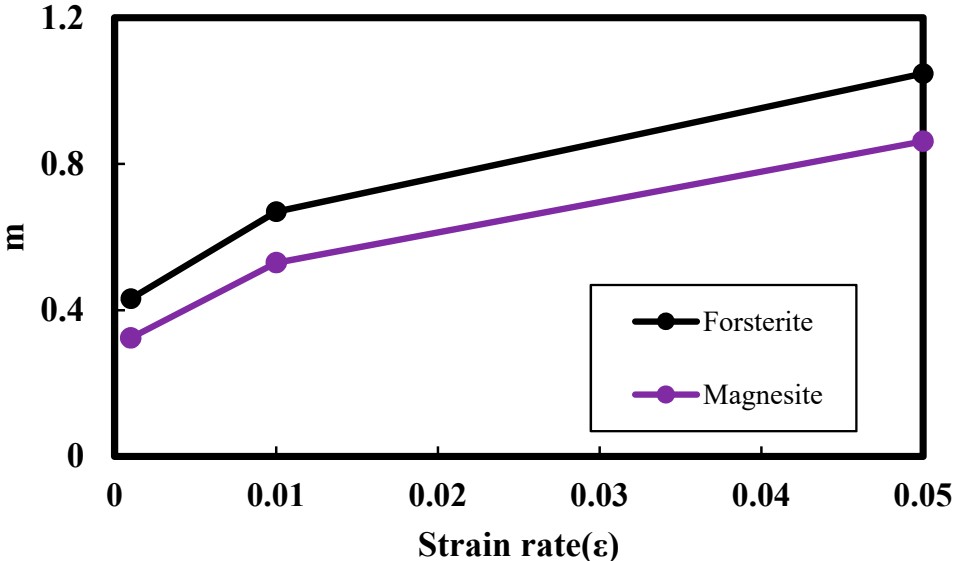

**Figure 12.** Strain rate sensitivity of both forsterite and magnesite at 500 K.

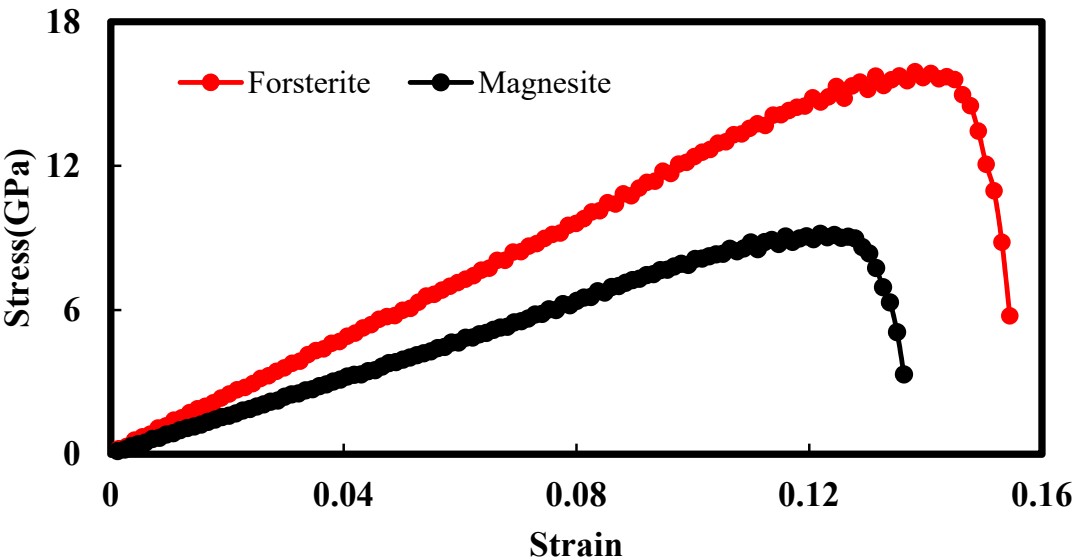

**Figure 13.** Compressive strength curve of forsterite and magnesite for lower grain sizes.

### 3.5. Elastic Properties

Elastic properties are crucial parameters of Earth's minerals utilized in geophysics and geochemistry to understand and explain the interior information of the Earth. These properties are indispensable for characterizing the rheology of geotectonics and constructing seismic acoustic waves in solid-state physics. The elastic properties represent the ability of materials to deform for a small change in stress. The crystal structure, composition, and microstructure of any material can be illustrated from these properties. Based on these properties, compositional changes, as well as elongations in materials, can be observed.

The elastic constant is the ratio of the second derivative of energy density concerning the strain and can be expressed simply as in Equation (5) [42].

$$C_{ij} = \frac{1}{V} \frac{\partial^2 E}{\partial \varepsilon_i \partial \varepsilon_j} \tag{5}$$

where $C_{ij}$ is the stiffness coefficient, $V$ is the volume of the unit cell, $E$ is the energy, and $\varepsilon$ is the strain.

Here, for calculating the elastic constants, a three-dimensional fourth-order Voigt notion model was used which considers the basis of Hooke's law, finite element analysis, and diffusion MRI [42]. This model is convenient for providing elastic constants of a material as a 6-by-6 matrix based on pressure tensors. Here, only three pressure- and temperature-dependent elastic stiffness constants ($C_{11}$, $C_{33}$, and $C_{44}$) for both forsterite and magnesite polycrystals have been considered to study the mechanical deformation of the minerals. These $C_{11}$, $C_{33}$, and $C_{44}$ coefficients are known as the pressure tensors (part of the 6-by-6 matrix) and represent the specific components of the stress and strain relationship. $C_{11}$ represents the elastic stiffness in the direction parallel to x axis, $C_{33}$ in the direction parallel to z-axis, followed by $C_{44}$ in the y–z plane (shear stiffness).

Figures 14 and 15 show the pressure- and temperature-dependent elastic constants for both forsterite and magnesite polycrystals. The obtained results agree well with those of previous works. For both forsterite and magnesite, the elastic constant values of $C_{11}$, $C_{33}$, and $C_{44}$ at 1 GPa and 300 K are validated with results from previous works in Table 6 [21,43]. As can be seen in Figures 14 and 15, the pressure and temperature changes noticeably affect the elastic constants of the two polycrystals. For forsterite (Figure 14a), the changes in pressure from 1 GPa to 100 GPa increased the values of $C_{11}$, $C_{33}$, and $C_{44}$ to 344.73, 620.59, and 859.89 GPa, respectively. On the other hand, these values were found to be approximately 220.61, 347.25, and 462.89 GPa, respectively, for magnesite (Figure 15a). So, it can be observed that the elasticity is linearly dependent on the applied pressure. At higher pressures, both forsterite and magnesite undergo a phase transition to a denser crystal structure. Also, Young's modulus values for these minerals are higher. Further, the higher pressure changes the bonding between the atoms. These phenomena cause the elastic properties of the minerals to be higher. In short, higher pressure and elasticity result in higher tensile strength for the two minerals.

However, increasing temperature increases the atomic vibration through the crystal structure and continues until it reaches a value where the atomic bonds become weak. This causes disorder and irregularity in the crystal lattice. As a result, the elastic properties of the minerals decrease and the minerals become less resistant to deformation. Hence, the temperature increasing from 300 K to 700 K decreases the values of $C_{11}$, $C_{33}$, and $C_{44}$ for forsterite by 35.94%, 45.33%, and 49.10% (Figure 14b) and for magnesite by 41.98%, 30.76%, and 47.09% (Figure 15b), respectively. Consequently, the temperature is reversely correlated to the elasticity of the materials. The reason is at the higher temperature, the higher the thermal vibration the materials have to go through which increases thermal expansion inside the materials but reduces the tensile strength and corresponding lattice constants. The study of these elastic properties for different pressures and temperatures is essential for investigating the phase transition of minerals (including mineralogy and geology). This phase transition is very helpful in interpreting the geophysical data and design of the

studied minerals for their desired properties. This study can give insights into developing a model of the studied minerals for predicting behaviors under extreme conditions.

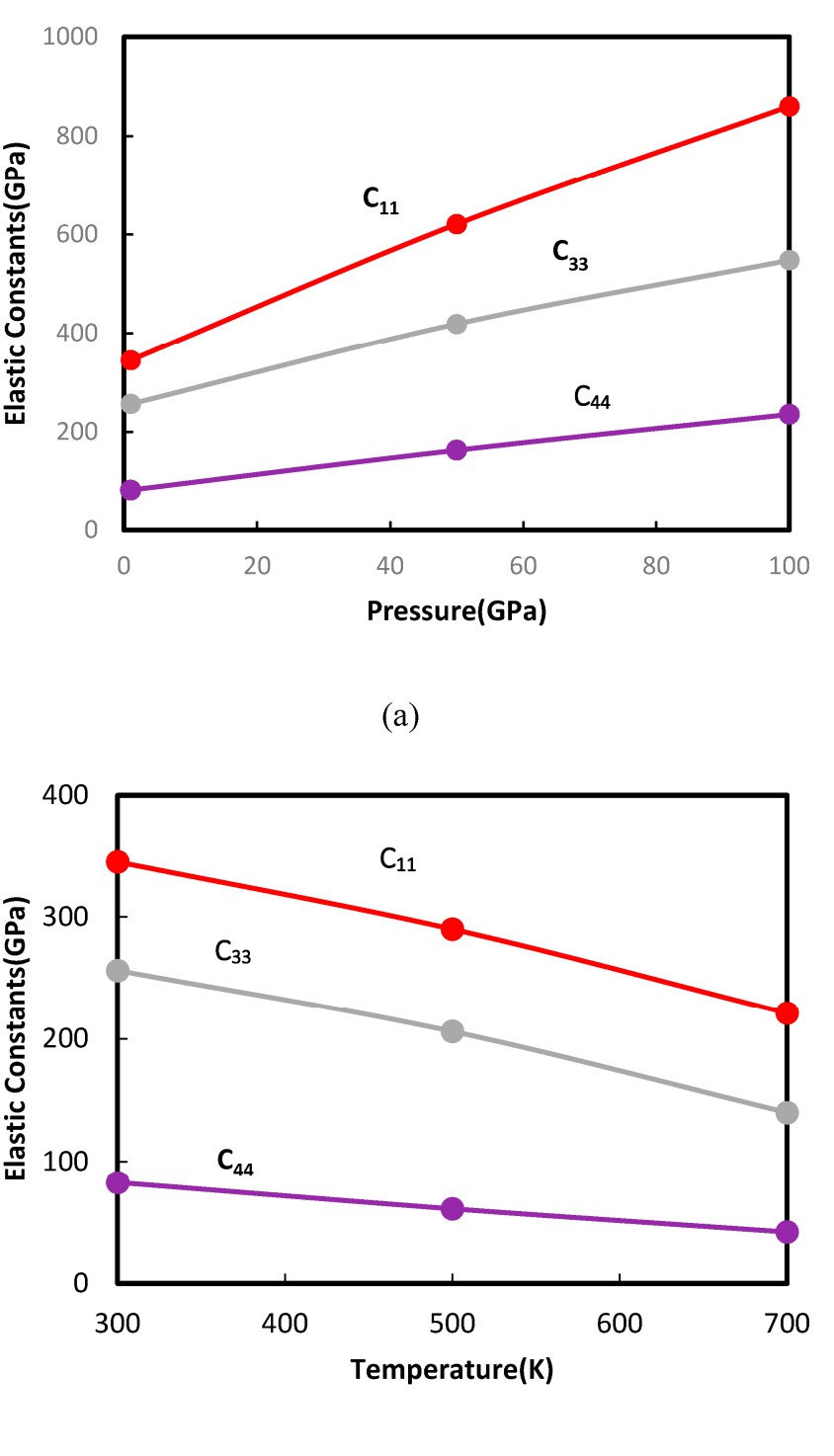

(a)

(b)

**Figure 14.** Elastic constants of forsterite: (**a**) effect of pressure and (**b**) effect of temperature.

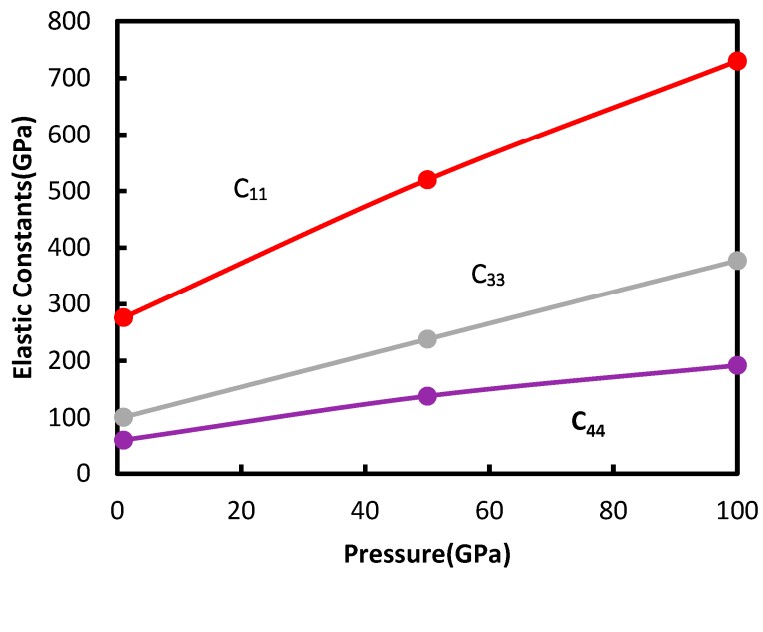

(a)

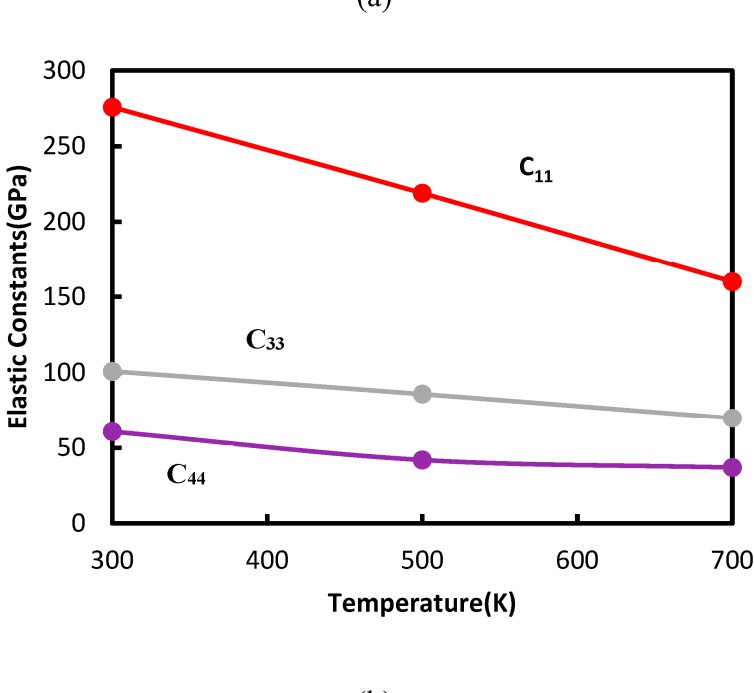

(b)

**Figure 15.** Elastic constants of magnesite: (**a**) effect of pressure and (**b**) effect of temperature.

**Table 6.** Elastic stiffness coefficients at 300 K.

| | Forsterite | | | Magnesite | | |
|---|---|---|---|---|---|---|
| | $C_{11}$ (GPa) | $C_{33}$ (GPa) | $C_{44}$ (GPa) | $C_{11}$ (GPa) | $C_{33}$ (GPa) | $C_{44}$ (GPa) |
| MD work (current study) | 345.0 | 256.1 | 82.52 | 275.85 | 101.79 | 60.81 |
| Experimental work [21,41] | 342.0 | 253.1 | 79.49 | 272.38 | 102.55 | 58.81 |

### 3.6. Radial Distribution Function

The radial distribution function (RDF) of any bonded simulated system is calculated mainly to determine the pairwise interaction and coordination number between groups

of nearest-neighbor atoms. The RDF represents the probability of finding a particle at a specific distance ($r$) from a reference particle in the mineral. Particularly, this function determines the distribution of neighboring particles around a center particle. The RDF for any bonded system can be expressed as follows:

$$g(r) = \frac{\rho(r)}{\rho} \tag{6}$$

where $\rho(r)$ is the average local number density of particles at a distance $r$, and $\rho$ is the bulk density of the particles.

The histogram form represents the RDF calculation by binning pairwise interactions into a distance of several bins. This distance is specified as the pair cutoff distance $r$ for a particular potential field. The RDF is counted only for the specified cutoff distance, and the coordination of any atoms beyond this distance is out of consideration. The RDF function shows better results (with multiple peaks in the histogram) if the system is uniform and well equilibrated. If the system is neither uniform nor well equilibrated, there is a sharp change in the coordination of the atoms with one single peak due to less interaction.

Here, Figure 16 shows the pair of RDFs for the disordered structures of both forsterite and magnesite polycrystals. Only the pairwise interactions between Si–O and C–O atoms are considered for this RDF calculation since the bonded interactions of these atoms are the most contributing factors for providing strength in the respective polycrystals. Even though several peaks can be seen in the figure, only the sharp peak is considered the strongest interaction between the pairs of atoms. The red curve is for the Si–O atoms whereas the black curve is for C–O atoms. These two RDF curves are obtained from the trajectories of the coordination numbers of their bonded atoms. The highest peak of the first curve is found around 2.54 angstrom which indicates the strongest interaction between the bonded Si–O atoms. On the other hand, the highest peak position is seen at 3.26 angstrom for bonded C–O atoms. In addition, the densities of both bonded Si–O and C–O atoms are higher at those peak points of the curves. These two curves' peak intensity indicates that the oxygen atoms usually attract the Si atoms more strongly than the C atoms. However, from the RDF calculation for these two polycrystals, it is observed that the weaker interaction between the carbon and oxygen atoms compared to the silicon and oxygen atoms makes magnesite less strong than forsterite.

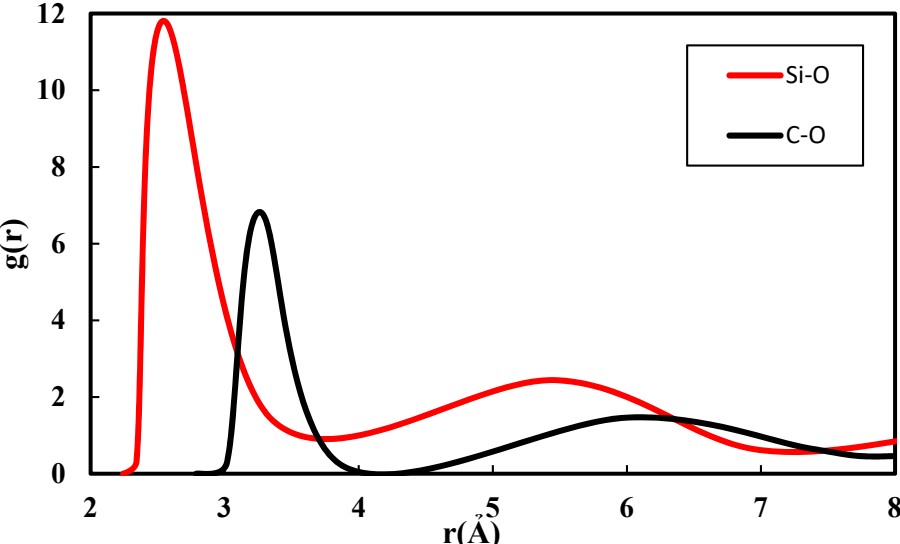

**Figure 16.** Pairwise Radial Distribution Function of forsterite and magnesite samples.

## 4. Conclusions

At the microscale, the presence of many mineral phases as well as microstructural defects distinguishes geomaterials, which are known to be heterogeneous and discontinuous. In summary, this study examines the changes in mechanical properties of forsterite and magnesite polycrystals using molecular dynamics simulation under different conditions. A direct comparison can be made between each computation as the MD simulation allows for the splitting of energy data calculations into individual parts such as surface energy, elastic energy, and plastic energy. In this work, the effects of strain rate and temperature on the stress–strain properties of the polycrystal models are studied.

The highlights of this research are listed below:

– The Young's modulus values of forsterite and magnesite are found to be approximately 154.7451 GPa and 92.84 GPa under tensile force, while these values are found to be around 120.457 GPa (forsterite) and 77.04 GPa (magnesite) under compressive force. Increasing temperature reduces the maximum strength of the polycrystals. For higher temperatures, forsterite shows higher ductility than magnesite;

– Higher strain rates require higher strain energy to initiate plastic deformation in the polycrystals. This effect is the opposite for the case of increasing temperature;

– According to the strain rate sensitivity results, magnesite shows less sensitivity to applied force than forsterite. At 300 K and a strain rate of $0.01 \text{ ps}^{-1}$, the SRS values of forsterite and magnesite are found to be approximately 0.70764 and 0.5716, respectively. The results also state that the impact of temperatures on SRS is higher for lower strain rates;

– Decreasing grain size (or increasing numbers of atoms) reduces the mechanical strength properties of the polycrystals. The yield strengths of the forsterite and magnesite dropped by 7.89% and 9.09% compared to the initial systems;

– Increasing pressure induces phase transition and increases the elastic properties of the polycrystals. On the other hand, increasing the temperature increases the atomic vibration through the crystal structure and this causes disorder and irregularity in the crystal lattice;

– In addition, from the RDF results, it is observed that the peak intensity of pairwise interaction between Si–O is higher than that for Mg–O.

Finally, this study has found that magnesite, which is the product of mineral carbonation of forsterite, is a favorable rock type for comminution. Magnesite shows less ductility at higher temperatures compared to forsterite. Our results imply that mineral carbonation impacts the energy requirements of minerals for comminution and serves as an energy-saving approach for mineral processing in addition to its effect on reducing greenhouse gases in the atmosphere.

**Author Contributions:** A.T.: methodology, investigation, validation, writing—original draft. B.N.: conceptualization, supervision, writing—review and editing. All authors have read and agreed to the published version of the manuscript.

**Funding:** This research received no external funding.

**Institutional Review Board Statement:** Not applicable.

**Informed Consent Statement:** Not applicable.

**Data Availability Statement:** Data will be made available upon request.

**Conflicts of Interest:** The authors declare no conflict of interest.

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
