# Peer review of "Molecular Dynamics Simulation of Forsterite and Magnesite Mechanical Properties: Does Mineral Carbonation Reduce Comminution Energy?"

_sustainability, doi:10.3390/su151612156_

Round 1

Reviewer 1 Report

The topic is interesting and it could be an important work. However, the organization of this paper is not appropriate, and the technical contents need significant improvement. Some comments are as follows:

1)    The main contribution of this study: molecular dynamics simulation (MDS), lacks of sufficient explanation about methodology, the specific consideration in this study and modeling details.

2)    The CO2 emission is a very weak point according to the contents of this paper; therefore, the title is not proper. This paper mainly discussed mechanical properties.

3)    All the presented stress-strain curves are simulated curves? If so, there are two problems: 1) Where are the simulation details of MDS? 2) The experimental verification is needed.

4)    The compressive/tensile strength can be xx GPa? Usually xx MPa.

5)    The layout of figures should be improved. Current quality is low.

6)    It would be very appreciated if some references can be reviewed, such as: Cold Regions Science and Technology 160 (2019) 31-38. Cold Regions Science and Technology 201 (2022) 103619. Cold Regions Science and Technology 201 (2022) 103621

None

Author Response

Comment 1: “The main contribution of this study: molecular dynamics simulation (MDS), lacks of sufficient explanation about methodology, the specific consideration in this study, and modeling details.”

  • The contribution of this study was to investigate the effect of different temperatures, pressures, loading rates, and grain size on the strength and elastic properties of both forsterite and magnesite at the molecular level. The thermal effects on the evolution of stress-strain properties for those two minerals are still not found in previous studies which can predict the stability and changes of surface conditions under higher temperatures and pressures during the comminution process (mentioned in the introduction part). In the first part of the methodological section, consideration of the MD simulation method for this study was described. In addition, materials and atomic model properties for the simulation, the selection of Buckingham potential, and all of the necessary forcefield parameters are mentioned in Tables 2 and 3.

We added the following to the manuscript:

  • Here, Moltemplate was used to generate the all-atom molecular models for the two geomaterials. The crystal systems for forsterite and magnesite were considered orthorhombic and trigonal, respectively, and the crystal orientations with specified lattice constants were made using the Voronoi algorithm method. The volumes of the simulation models for forsterite and magnesite are 125nm3 and 117.65nm3. The Verlet algorithm was used to calculate the dynamic trajectories of the particles. The Ewald summation method was used to calculate the long-range coulombic interaction among the atoms. A Noose-Hoover thermostat and barostat were used to control the temperature and pressure during the simulation.  Also, the simulation procedures, simulation time, and minimization method are mentioned in the last part of the methodological section. Lastly, the respective equations for calculating the stress-strain values, elastic stiffness, and radial distribution function are described with references in each section of this paper.  This radial distribution function determines the pairwise interactions and coordination number between the groups of neighbor atoms for any bonded simulated system. The peak intensities of the pairwise interactions between Si-O and Mg-O were calculated to show the histogram of the radial distribution functions of forsterite and magnesite. Besides, the equilibrium procedures are also described in the methodological part.

Comment 2: “The CO2 emission is a very weak point according to the contents of this paper; therefore, the title is not proper. This paper mainly discussed mechanical properties.”

  • Although we do not simulate the rock reactions with CO2 here, the main focus of this work is to answer the question whether the physical and mechanical properties of silicates after reacting with CO2 (which is known as mineral carbonation) is changed and whether this change is favorable in terms of minimizing the energy requirement for the comminution process. So we still believe we should keep the terms carbonation and comminution in the title, but we modified it to put more focus on the simulation of mechanical properties of these two rocks.

  • Comment 3: “All the presented stress-strain curves are simulated curves? If so, there are two problems: 1) Where are the simulation details of MDS? 2) Th experimental verification is needed.”
  • Yes, all of the presented curves are generated from the MD simulation. The simulation details as well as the necessary equations are described in the methodological sections and each results section also. For example, the selection of temperatures, pressures, loading rates, and choice of the system axis to make deformation, all are mentioned in the methodological part. The validation of the works is added in the paper in Figure 3. Such as Figures 3b and 3c show the comparison of the stress-strain curves between the current study and the previous study for both materials. And, the comparisons of the elastic constants for both models are provided in Table 6. Thanks for notifying me about this important issue.

Comment 4: “The compressive/tensile strength can be xx GPa? Usually xx MPa.”

  • The reason for considering GPa as a unit for strength values over MPa was for comparing the results with previous works. The previous works used the GPa unit for their results.

Comment 5: “The layout of figures should be improved. Current quality is low”

  • Checked and replaced two figures (7a and 14b) with good quality for better presentation in the paper.

Comment 6: “It would be very appreciated if some references can be reviewed, such as: Cold Regions Science and Technology 160 (2019) 31-38. Cold Regions Science and Technology 201 (2022) 103619. Cold Regions Science and Technology 201 (2022) 103621”

  • These papers are reviewed thoroughly and cited in the resultant part (effect of grain size) for their similar application in particle size distribution in my work.

Reviewer 2 Report

The manuscript titled "Molecular Dynamics Simulation of Forsterite and Magnesite Mechanical Properties: Effect of Carbonation on Comminution Energy" can be accepted for publication after its minor revision. There are some my suggestions for its revision. 

1. Minor editing of English language required.

2. In some figures, the axis writings were missed (Figure  and Figure 6a). Please check and complete them. 

The manuscript titled "Molecular Dynamics Simulation of Forsterite and Magnesite Mechanical Properties: Effect of Carbonation on Comminution Energy" can be accepted for publication after its minor revision. There are some my suggestions for its revision. 

1. Minor editing of English language required.

2. In some figures, the axis writings were missed (Figure  and Figure 6a). Please check and complete them. 

Author Response

We thank all the reviewers for their feedback. Please find below our response to each question. We used track changes to address the reviewers’ comments and revised the manuscript accordingly.

Comment 1: “Minor editing of English language required.”

  • The whole manuscript is revised thoroughly and made some corrections on grammatical errors and sentence structures where needed.

Comment 2: “In some figures, the axis writings were missed (Figure 6a). Please check and complete them.”

  • The axis titles are given in Figure 7a (the figure number 6 is changed to figure number 7). Thanks for notifying me about this error.

Reviewer 3 Report

The manuscript requires a number of corrections before any decision on the manuscript.

1. There are a number of Typo errors. Authors are suggested to remove them.

2. See graph 13 (b) and again draw the graph as there is a discontinuity on Y-axis.

3. Some latest references are missing from the text. Authors are suggested to include the latest references in the manuscript.

4. Introduction is too lengthy. Do not make it lengthy unnecessarily. Authors are suggested to modify the Introduction and make it more presentable discussing the core of the research topic only.

The language can be improved further.

Author Response

We thank all the reviewers for their feedback. Please find below our response to each question. We used track changes to address the reviewers’ comments and revised the manuscript accordingly.

Comment 1: “There are a number of Typo errors. Authors are suggested to remove them.”

  • The manuscript is thoroughly revised and made corrections to the errors where needed.

Comment 2: “See graph 13 (b) and again draw the graph as there is a discontinuity on Y-axis.”

  • I have replaced the old figure with the new layout where there is no discontinuity can be found. To be noted, figure 13 is changed to 14.

Comment 3: “Some latest references are missing from the text. Authors are suggested to include the latest references in the manuscript.”

  • Few recent works on studying the changes in mechanical properties of geomaterials using MD simulation are cited and added (effect of grain size).

Comment 4: “Introduction is too lengthy. Do not make it lengthy unnecessarily. Authors are suggested to modify the Introduction and make it more presentable discussing the core of the research topic only.”

  • I have reduced the length of the introduction part with much focusing on the core of the research topic as much as possible. The basic introductory information about the mineral carbonation, comminution process, CO2 emissions, and reutilizing of carbon products are brought into short discussion within limited sentences.

Round 2

Reviewer 1 Report

The authors have modified the paper accordingly, I have no further comments.

None